# Effects of Oral Function Training and Oral Health Status on Physical Performance in Potentially Dependent Older Adults

**DOI:** 10.3390/ijerph182111348

**Published:** 2021-10-28

**Authors:** Masataka Sasajima, Akihiro Yoshihara, Ayuko Odajima

**Affiliations:** 1Health Promotion Division, Department of Health and Social Welfare, Niigata Prefectural Government, 4-1 Shinko-cho, Chuo-Ku, Niigata 950-8570, Japan; sasajima.masataka@pref.niigata.lg.jp; 2Division of Oral Science for Health Promotion, Faculty of Dentistry & Graduate School of Medical and Dental Science, Niigata University, 2-5274 Gakkocho-dori, Chuo-Ku, Niigata 951-8514, Japan; ayuko@dent.niigata-u.ac.jp

**Keywords:** physical function, oral function training, balance ability, oral diadochokinesis

## Abstract

This study aimed to evaluate the effects of an oral function training program and indicators of oral health status on improvements in physical performance induced by physical function training in dependent older adults. The participants were 131 potentially dependent older adults (age: ≥65 years) who were randomly divided into two groups: an oral intervention and a control group. All participants underwent physical function training, but only the intervention group took part in the oral function training program. In total, 106 participants completed all of the survey components (60 and 46 participants from the intervention and control groups, respectively). The measures of physical fitness examined included the one-leg standing time with eyes open (OLST) and the timed up and go test (TUG). Logistic regression analyses were carried out to determine the effects of the oral function intervention and health status on physical fitness. The results revealed that the oral function intervention significantly improved OLST and TUG times. These findings suggest that evaluations of oral health status and interventions aimed at activating oral functions are associated with improvements in physical fitness among potentially dependent older adults.

## 1. Introduction

Increases in life expectancy have led to a rise in the number of older people in Japan. In 2014, 26.0% of the Japanese population was ≥65 years of age [1]. As a result of these changes, the number of older people in need of support or care has also increased, which has led to the increased need for interventions aimed at preventing reductions in people’s ability to perform activities of daily living (ADLs). ADLs are considered to be a major component of quality of life (QOL). It has been shown that the ability to perform ADLs is adversely affected by functional impairments, frailty, and falls, and these factors are significantly influenced by physical performance parameters such as muscle strength and balance function [2]. In 2006, a decision was made to establish a long-term care project aimed at preventing reductions in the ability to perform ADLs and delaying the deterioration of such abilities to the point at which care would be required [3]. The target of this project was high-risk older people, defined as older individuals whose health was deemed to be at high risk of deteriorating to the point at which support or care would be required. The preservation and improvement of physical and oral functions were introduced as components of this long-term care project. According to a meta-analysis by 45 articles, low one-leg standing time with eyes open (OLST), timed up and go (TUG), and chair stand test time were associated with worsening ADL. Muscle measures at baseline are predictors of future ADL and IADL dependence in the older adult population [4].

On the other hand, according to previous studies [5,6], several oral health status parameters influence physical functions. For example, it has been suggested that dental occlusal support and masticatory performance are related to leg extensor power and balance function in older people, as measured by OLST. Izuno et al. [7] reported that tongue and lip movements, including oral diadochokinesis (OD), are associated with physical fitness.

In addition, a previous study that examined the effects of oral function improvement programs [8] reported that an oral function intervention for older individuals was useful at improving oral functions such as OD and repetitive saliva swallowing test scores. Furthermore, Ibayashi et al. [9] reported that an oral exercise program improved the biting force, swallowing ability, and salivary flow rate in older people, particularly those who had ≥20 teeth remaining. In recent years, good nutrition is considered the basis for achieving a good quality of life. A good diet, together with constant physical activity, can increase not only the quality of life but also the life expectancy of people [10].

However, the enforcement rate of oral function improvement programs is lower than that of physical function improvement programs because the benefits of oral function improvement programs are difficult for caregivers and those in charge of health in local communities to understand. To our knowledge, there is not the studies that have evaluated the effects of combining physical function training with oral function training. It is hypothesized that the combination of oral function training to general physical training might give bigger improvement of not only oral function but also the physical performance. Therefore, this study aimed to evaluate the effects of an oral function training program on the improvement in physical performance induced by physical function training in older individuals.

## 2. Methods

### 2.1. Study Participants

This study was a cross-sectional study. All of the participants in this study were high-risk older individuals (age ≥ 65 years), defined by the Ministry of Health, Labour and Welfare (MHLW) in Japan as older individuals whose health was deemed to be at high risk of deteriorating to the point at which support or care would be required. In addition, all participants were taking part in a preventive, long-term, functional improvement program targeting the musculoskeletal system. All of the participants were living in one of two residential areas: Sado city and Yahiko village (both in Niigata Prefecture, Japan). 

All participants completed a basic health checklist (Figure 1) that was based on the preventive long-term care manual published by the MHLW. We conducted personal interviews established by the MHLW to obtain information regarding habitual function, physical function, nutritional status, social withdrawal, dementia, and depression, in addition to oral status such as chewing function, feelings of suffocation while eating, and dry mouth. Individuals who met three or more of items 6–10 on the checklist or were selected by the local government were considered to be eligible for participation in this study. A total of 180 high-risk older people were selected according to the above procedures, among whom, 131 chose to participate. All participants were randomly divided into two groups—an oral intervention group and a control group—without consideration of their physical or oral function or their wishes before completing the baseline survey. A total of 106 people (22 males and 84 females) participated in the follow-up survey, which was performed 3 months after the baseline survey, and subjected to analysis (Figure 2). Written informed consent was obtained from all participants before the study began after being fully informed of the study purpose and methods. This study was approved by the ethics committee of the Faculty of Dentistry, Niigata University (24-R23-02-06).

### 2.2. Survey Methods

A physical function improvement program based on the preventative long-term care manual published by the MHLW was used in this study [11]. The program was performed for 1.5 h a week for 3 months. Specifically, physical therapists instructed the participants to perform physical exercises aimed at enhancing their physical functions, such as their balance ability, walking ability, and ability to perform ADLs [11].

The program consisted of three phases: a conditioning phase, a strengthening phase, and a functional training phase. The conditioning phase was designed to enable the participants to become accustomed to performing physical functions through low-intensity high-repetition exercises. In this phase, the participants mainly learned about speed control and training forms. The next phase, the strengthening phase, started when the participants were able to perform the exercise movements smoothly. In this phase, the weights used were adjusted to ensure that the intensity of the exercises was moderate to high, and the load strength was increased if the participants could complete the program effortlessly. The last phase, the functional training phase, involved more functional exercises based on ADLs. Balance training such as stepping and walking exercises were implemented in addition to the strength training. Each exercise mainly aimed to improve the muscular strength of the lower limbs and consisted of two or three sets of 10 repetitions. This program also involved warm-up and cool-down periods of 20 min that aimed to stretch the muscles.

### 2.3. Evaluation of Physical Function

Two physical fitness tests were performed, at baseline and after the intervention. The OLST was measured to evaluate static balance function [12]. In the OLST test, the participants were asked to stand on one leg with their eyes open and their arms outstretched until they lost their balance and had to stand on both legs. The maximum value for the right or left leg was used as the value for each participant, up to a maximum of 120 s. Values that exceeded 120 s were recorded as 120 s. The OLST test was performed twice, and the mean time was recorded. The timed up and go test (TUG), which was originally developed as a clinical measure of functional mobility in older people [13], measures the time taken to stand up from an armchair, walk a distance of 3 m, turn, walk back to the chair, and sit down. The TUG was also performed twice, and the mean time was recorded. We evaluated the normality of the distribution of the TUG by skewness and the kurtosis test for normality (*p* > 0.05).

### 2.4. Oral Function Improvement Program

Of the 106 participants, 60 belonged to the oral function intervention group, which performed orofacial myofunctional exercises. This program aimed to improve the participants’ oral function and was performed for 30 min a week for 3 months. Dental hygienists instructed the participants to perform four kinds of orofacial exercises (Figure 3).

### 2.5. Evaluation of Oral Health Status

Two trained dentists evaluated OD and the number of remaining teeth in both the oral intervention and control groups at baseline and the 3-month follow-up examination. OD is a measure of orofacial motor skills [14,15]. For the assessment of OD, the participants were asked to articulate the /pa/ syllable repeatedly as quickly as possible for 5 s, and the number of articulations was counted. The same procedure was repeated for the syllables /ta/ and /ka/, and the number of articulations was counted using a digital counter (Takei Scientific Instruments Co., Ltd., Niigata, Japan). OD scores were calculated separately for the /pa/, /ta/, and /ka/ syllables and recorded as the mean number of articulations per second. The ability to repeat the syllable /pa/ is a measure of lip function. On the other hand, the ability to repeat the syllables /ta/ and /ka/ is indicative of tongue root motor skills.

### 2.6. Characteristics of the Study Participants at Baseline

The following items (all of which were considered to be factors that might influence physical fitness) were assessed at baseline: age, gender, body mass index (calculated as the height in meters divided by the weight in kilograms squared), and residential area (Yahiko village or Sado city).

### 2.7. Statistical Analysis

The physical fitness measurements and OD scores were graded on a scale of 0–3 (with lower ranks indicating lower scores for each test). For each parameter, the participants were divided into quartiles at baseline. The participants were also divided using the same quartile cutoff values after the intervention.

The participants in ranks 0 to 2 were examined for a deterioration of 1 rank or more. Rank 0 can deteriorate to any of the ranks from 1 to 3, as well as from rank 1 to ranks 2 or 3. In addition, rank 2 can deteriorate to rank 3. On the other hand, during the analyses of the effects of each OD score, the participants that exhibited a rank of 0 or 1 at baseline were defined as the low score group, and those with ranks of 2 or 3 were defined as the high score group. In addition, during the analysis of the effects of the number of remaining teeth, the participants who had ≥20 teeth remaining at baseline were defined as the high score group, and the others were defined as the low score group [16]. The frequency of improvement in physical fitness was assessed according to oral health status at baseline using the χ^2^-test. In addition, logistic regression analyses were carried out to determine the effects of oral health status at baseline and of the oral function intervention on the improvement in physical fitness induced by the physical training program after adjusting for confounding variables. The physical fitness measurements (OLST and TUG) were selected as the dependent variables. On the other hand, the group (intervention or control) was selected as the explanatory variable. Oral health status was also selected. Two logistic models were constructed for each dependent variable. In Model 1, the oral function training intervention was selected as an explanatory variable. Similarly, the number of remaining teeth and OD scores were chosen as explanatory variables in Model 2. In addition, gender and age were used as covariates in each model. All statistical analyses were performed using STATA version 16.0 for Windows (StataCorp LP, College Station, TX, USA). Statistical significance was set at *p* < 0.05.

## 3. Results

Table 1 shows the characteristics of the study participants at baseline. No significant differences were found between the intervention and control groups (*p* > 0.05, χ^2^-test or Student’s *t*-test).

The physical fitness and oral health status data were obtained at the baseline and follow-up examinations (Table 2). During the comparisons of each physical fitness or oral health parameter between the baseline and follow-up examinations, mean values were used for the OLST, TUG, and OD data. The OLST data are shown as median (25th/75th percentiles) values because they were not normally distributed.

Significant differences in the oral intervention group were detected between the baseline and follow-up data. Regarding the physical fitness measurements, the median OLST improved significantly, from 16.5 s (2.8/27.6) to 24.1 s (9.1/36.2) (*p* < 0.05, Wilcoxon’s signed-rank test), as did the mean TUG time, from 9.1 s (±1.7) to 8.0 s (±1.3) (*p* < 0.05, paired *t*-test). In addition, all OD scores improved significantly after the intervention (*p* < 0.05, paired *t*-test). On the other hand, only the TUG time improved significantly (*p* < 0.05, paired *t*-test) after the intervention in the control group.

The quartile cutoff values for the OLST and TUG times are shown in Table 3. Based on these cutoff values, the improvements in physical fitness seen in each oral function group are shown in Table 4. Improvements in the OLST were significantly associated with the oral intervention and oral health status: oral function training program (*p* = 0.042) and number of remaining teeth (*p* = 0.046). On the other hand, improvements in the TUG time were associated with the oral function intervention program (*p* = 0.010).

In addition, the oral function intervention program had a significant positive effect on the physical fitness parameters (OLST: odds ratio [OR]=4.31, *p* = 0.027 for Model 1 and 6.48, *p* = 0.023 for Model 2; TUG: OR = 9.66, *p* = 0.004 for Model 1 and 13.90, *p* = 0.011 for Model 2). On the other hand, the number of remaining teeth was associated only with improvements in the OLST (OR = 7.44, *p* = 0.012 for Model 2). OD [/pa/] score was significantly associated with both the OLST (OR = 8.75, *p* = 0.028 for Model 2) and TUG (OR = 8.13, *p* = 0.044 for Model 2) by the logistic regression analysis (Table 5).

## 4. Discussion

In this intervention study, an oral function training program was found to have a significantly beneficial effect on physical fitness parameters. Indicators of oral health status, such as the number of remaining teeth and OD, were also shown to be related to such improvements. These results suggest that the activation and maintenance of oral function have a positive influence on physical function.

Tongue dexterity, as characterized by tongue movement from side-to-side, and OD were particularly associated with physical fitness [7]. OD is useful for measuring functional changes in utterances. The syllables /pa/, /ta/, and /ka/ were evaluated based on the backward function of the tongue, the function of the front of the tongue, and the function of the lips, respectively [16]. In the present study, we did not evaluate swallowing or mastication function. The risk of aspiration increases among individuals with OD. In addition, a significant difference was found in individuals with different /ta/ and /ka/ utterances, suggesting an association with motor ability during swallowing [17]. Therefore, it might be reasonable to use OD as a representative index of oral function [15].

The OLST and TUG times were selected as physical parameters in this study. The OLST is used to measure balance function because it is easy to administer and is associated with lower limb muscle strength. A reduction in the OLST could be a risk factor for falling and a reduced ability to perform ADLs [18,19]. In this study, the number of remaining teeth, OD scores (/pa/, /ta/, and /ka/), and the oral function training intervention were all found to be significantly related to improvements in the OLST. In previous studies [5,6], the number of remaining teeth, dental occlusion, and masticatory performance were shown to be associated with the OLST. In the present study, the number of remaining teeth was associated with improvements in the OLST, which agrees with the findings of previous studies. On the other hand, tongue movement skills and OD were shown to be related to the OLST in a previous study [7], and oral function training aims to activate orofacial myofunctions, including in the tongue.

It has been suggested that balance function is influenced by orofacial sensory inputs, such as dental occlusion, tongue movement, and lip motion, via the trigeminal nerve [20,21,22]. As dental occlusion might be associated with physical function, it is possible that these orofacial sensory inputs influence the neural control of muscle strength in other parts of the body. It has been reported that increasing tongue dexterity and movement leads to increases in orofacial sensory inputs [23]. These inputs affect the stabilization of head posture and body trunk movements. In agreement with this, an improvement in balance function was observed in the intervention group in the present study.

The TUG is used to assess functional mobility, which incorporates limb muscle strength, walking ability, dynamic balance, and agility. The TUG is also used to predict falls and the ability to perform ADLs in older adults [7]. The tongue and lip are composed of numerous muscle fibers, and that the dexterity and agility of these fibers are associated with general functional mobility. In the present study, the OD score for the /pa/ syllable and the oral function training intervention were found to be significantly related to improvements in the TUG time; these results agree with those of previous studies [7].

Our findings suggest that maintaining and activating oral function, including retaining teeth, and the ability to perform tongue and lip movements, are associated with improvements in physical fitness. These findings might contribute to the maintenance of a high QOL and ability to perform ADLs among older adults.

The present study did have some limitations. First, we were not able to carry out detailed analysis such as improvement in physical fitness after the intervention due to lack of enough sample, and there was a lack of information about the subjects’ medical histories and diseases related to aging, according to the evidence based dentistry [24]. Second, we considered a variety of factors, such as workforce, when determining the study regions and number of participants. However, a priori sample size calculation was not performed for this study. Third, we were not able to carry it out to measure of swallowing and mastication functions because a specialized measure such as endoscope or x-ray was necessary. Finally, we did not evaluate the participants’ occlusal status or prosthetic condition. A more detailed analysis is needed to confirm the relationship between oral function and physical fitness.

## 5. Conclusions

We evaluated the effects of an oral function training program and indicators of oral health status on the improvement in physical performance induced by physical function training in the dependent elderly. All of the participants in this study were high-risk older individuals (age ≥ 65 years), defined by MHLW in Japan as older individuals whose health was deemed to be at high risk of deteriorating to the point at which support or care would be required. Evaluations of oral health status, and interventions aimed at activating oral functions, were found to be associated with improvements in physical fitness in potentially dependent elderly. The training programs might contribute to the maintenance of ADLs and QOL in older adults. Therefore, the wider implementation of oral function improvement programs could be beneficial. In future, a much more detailed study is needed to confirm the relationship between oral function and physical fitness.

## Figures and Tables

**Figure 1 ijerph-18-11348-f001:**
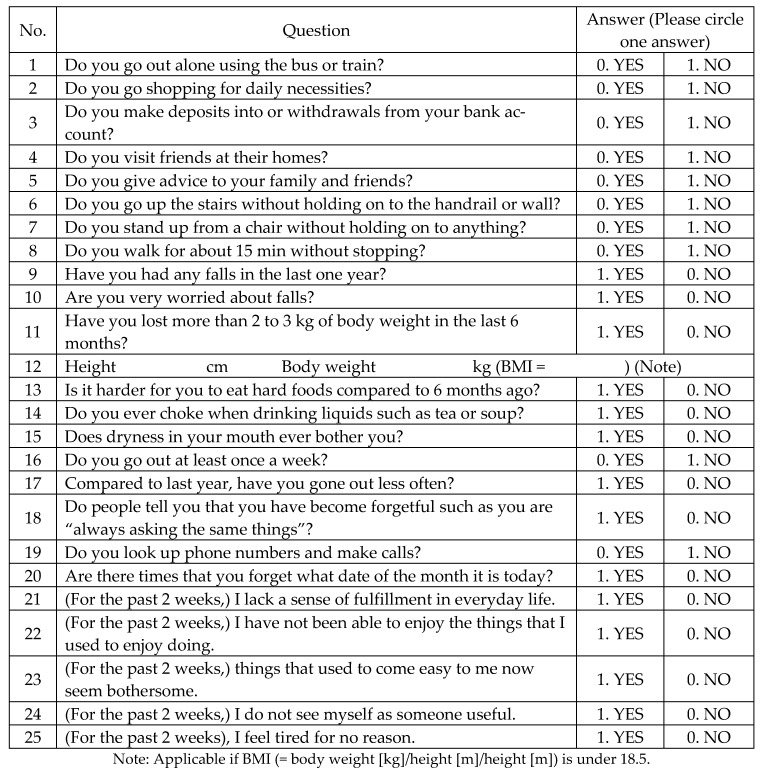
Basic health checklist (The Ministry of Health, Labour and Welfare of Japan Home Page. Available online: https://www.mhlw.go.jp/topics/2009/05/dl/tp0501-1c_0001.pdf (Japanese, accessed on 18 October 2021).

**Figure 2 ijerph-18-11348-f002:**
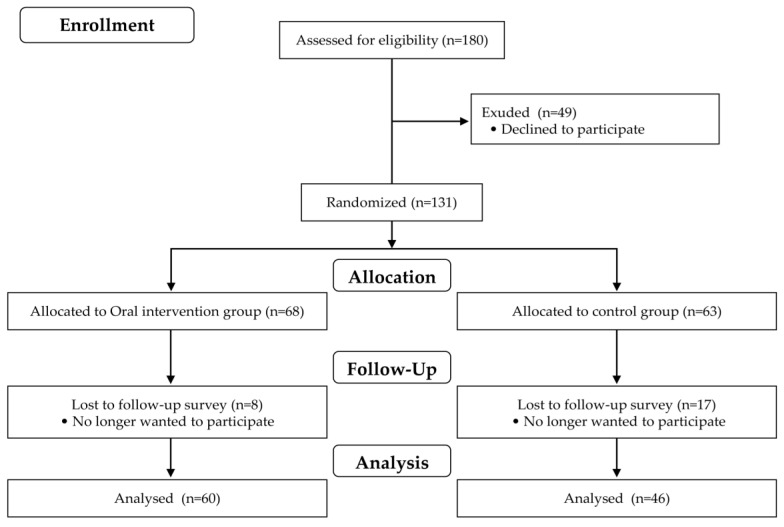
Flowchart of the study.

**Figure 3 ijerph-18-11348-f003:**
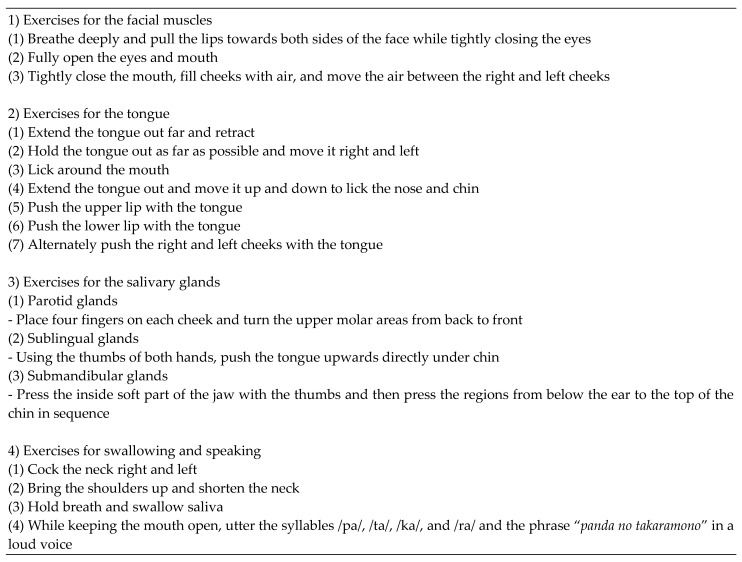
The oral function exercise program consisting of four exercises (exercises focusing on the facial muscles, tongue, salivary glands, and swallowing).

**Table 1 ijerph-18-11348-t001:** Selected characteristics of the participants in the intervention and control groups.

Number of Participants (%), mean ± SD			
	Group		*p*-Value
	Intervention Group	Control Group	
	*n* = 60	*n* = 46	
Age (years)			
≥80	21 (35.0)	14 (30.4)	0.620 *
<80	39 (65.0)	32 (69.6)	
Gender			
Male	12 (20.0)	10 (21.7)	0.827 *
Female	48 (80.0)	36 (78.3)	
Residential area			
Yahiko village	22 (36.7)	24 (52.2)	0.110 *
Sado city	38 (63.3)	22 (47.8)	
Dry mouth	27 (45.0)	30 (65.2)	0.313 *
Choking	26 (43.3)	26 (56.5)	0.661 *
Difficulty eating	26 (43.3)	21 (45.7)	0.657 *
Remaining teeth			
≥20	29 (48.3)	26 (56.5)	0.403 *
<20	31 (51.7)	20 (43.5)	
BMI	23.9 ± 4.0	24.6 ± 3.6	0.353 **

* χ^2^ test. ** Student’s *t*-test. BMI: body mass index.

**Table 2 ijerph-18-11348-t002:** Descriptive statistics for various physical fitness and oral function status measurements at the baseline and follow-up examinations.

Measurements	Intervention Group		Control Group	
	Baseline	Follow-Up	Baseline	Follow-Up
Physical fitness (s)				
OLST *	16.5 (2.8/27.6)	24.1 (9.1/36.2) ^a^	19.6 (4.0/22.0)	20.6 (5.1/29.6)
TUG **	9.1 ± 1.7	8.0 ± 1.3 ^a^	9.5 ± 2.8	8.5 ± 2.6 ^a^
Oral functional status (counts/s)				
OD (/pa/) **	6.1 ± 1.1	6.3 ± 0.8 ^a^	6.2 ± 0.7	6.2 ± 0.9
OD (/ta/) **	6.0 ± 0.8	6.2 ± 0.6 ^a^	6.2 ± 0.6	6.2 ± 0.5
OD (/ka/) **	5.7 ± 0.9	5.9 ± 0.7 ^a^	5.8 ± 0.9	6.0 ± 0.6

OLST: one-leg standing time with eyes open; TUG: timed up and go test; OD: oral diadochokinesis. * Median (25th/75th percentile). ** Mean ± SD. ^a^ Significant differences (*p* < 0.05) were found between the baseline and follow-up values according to Wilcoxon’s signed-rank or the paired *t*-test.

**Table 3 ijerph-18-11348-t003:** Quartile cutoff values for various physical fitness and oral functional status measurements at baseline (Min.-Max, N: number of participants).

Measurements	Rank 0	Rank 1	Rank 2	Rank 3
Physical fitness (s)				
OLST	5.5–7.6 N = 27	7.7–8.7 N = 26	8.8–10.5 N = 28	10.5–17.4 N = 25
TUG	0–3.2 N = 27	3.3–8.2 N = 26	9.3–24.0 N = 27	27.0–105.2 N = 26

OLST: one-leg standing time with eyes open; TUG: timed up and go test.

**Table 4 ijerph-18-11348-t004:** Improvements in physical fitness seen in each oral function group, n (%).

Measurements	Group	OLST		*p*-Value	TUG		*p*-Value
		Improvement (+)	Improvement (–)		Improvement (+)	Improvement (–)	
Oral health status (at the baseline)							
Remaining teeth	High score	14 (58.3)	10 (41.7)	0.046	14 (56.0)	11 (44.0)	0.103
	Low score	9 (31.0)	20 (69.0)		11 (35.5)	20 (64.5)	
OD (/pa/)	High score	9 (52.9)	8 (47.1)	0.252	7 (43.8)	9 (56.3)	0.586
	Low score	14 (38.9)	22 (61.1)		18 (45.0)	22 (55.0)	
OD (/ta/)	High score	7 (36.8)	12 (63.2)	0.569	6 (35.3)	11 (64.7)	0.263
	Low score	16 (47.1)	18 (52.9)		19 (48.7)	20 (51.3)	
OD (/ka/)	High score	9 (42.9)	12 (57.1)	0.588	10 (50.0)	10 (50.0)	0.374
	Low score	14 (43.8)	18 (56.3)		15 (41.7)	21 (58.3)	
Oral function program	Intervention	17 (54.8)	14 (45.2)	0.042	19 (59.4)	13 (40.6)	0.010
	Control	6 (27.3)	16 (72.7)		6 (25.0)	18 (75.0)	

**Table 5 ijerph-18-11348-t005:** Logistic regression analysis of each physical fitness parameter.

Independent Variables	Dependent Variables: OLST [1: Improvement (+)]					
	Model 1: Intervention			Model 2: Remaining Teeth and OD		
	OR	*p*-Value	95% CI	OR	*p*-Value	95% CI
Gender						
1: Female	1.12	0.881	0.26–4.80	1.49	0.656	0.26–8.52
Age						
	0.89	0.051	0.79–1.00	0.82	0.009	0.71–0.95
Model 1:						
Intervention						
Oral function training intervention	4.31	0.027	1.18–15.69	6.48	0.023	1.29–32.47
1: Intervention (+)						
Model 2:						
Remaining teeth and OD						
Remaining teeth						
1: ≥20				7.44	0.012	1.55–35.63
Oral diadochokinesis (/pa/)						
1: High score				8.75	0.028	1.27–60.24
Oral diadochokinesis (/ta/)						
1: High score				0.20	0.138	0.03–1.67
Oral diadochokinesis (/ka/)						
1: High score				0.89	0.90	0.15–5.43
Number of participants		53			53	
Coefficient of determination (R^2^)		0.114			0.277	
**Independent Variables**	**Dependent Variables: TUG [1: Improvement (+)]**					
	**Model 1: Intervention**			**Model 2: Remaining Teeth and OD**		
	**OR**	***p*-Value**	**95% CI**	**OR**	***p*-Value**	**95% CI**
Gender						
1: Female	1.34	0.723	0.27–6.76	1.80	0.548	0.26–12.21
Age						
	0.79	0.001	0.68–0.91	0.70	<0.001	0.58–0.86
Model 1:						
Oral function training intervention						
1: Intervention (+)	9.66	0.004	2.07–45.02	13.90	0.011	1.81–106.90
Model 2:						
Remaining teeth and OD						
Remaining teeth						
1: ≥20				4.53	0.073	0.87–23.64
Oral diadochokinesis (/pa/)						
1: High score				8.13	0.044	1.06–62.37
Oral diadochokinesis (/ta/)						
1: High score				0.17	0.138	0.02–1.77
Oral diadochokinesis (/ka/)						
1: High score				2.55	0.404	0.28–22.91
Number of participants		53			53	
Coefficient of determination (R^2^)		0.300			0.415	

## Data Availability

The data presented in this study are available on request from the corresponding author. The data are not publicly available due to privacy.

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
