# Peer review of "Effects of Oral Function Training and Oral Health Status on Physical Performance in Potentially Dependent Older Adults"

_ijerph, 2021, doi:10.3390/ijerph182111348_

Round 1

Reviewer 1 Report

Dear authors,

Thank you for presenting an interesting manuscript about "Effects of oral function training and oral health status on the 2 physical performance of the potentially dependent elderly". 

The present manuscript has merit for publication.

However, I want to suggest some modifications to improve your study.

  • Improve the English language.
  • Change the number "2" in the title per "two".
  • Improve the quality of your abstract.
  • Provide the type of the study and the power of the sample size in the Materials and Methods section.
  • Figure 1 - Include the reasons for patients' loss.
  • Did you consider the Skewness and kurtosis test for normality p>0.005? Is that correct?
  • I could not understand "Table 1" because it is not a table.
  • The authors should explore the data obtained and write that in the results section. This section should be informative with a straightforward interpretation for the readers.
  • Please, do not write only "p." Change it for "p-value."
  • Fix the Table 4.
  • Include the limitations of the present study in your discussion.

Regards,

#Reviewer

Author Response

Our point-by-point reply to the reviewer 1’s comments

We are grateful to reviewer 1 for the critical comments and useful suggestions that have helped us to improve our paper considerably. As indicated in the followings, we have taken all these comments and suggestions into account in the revised version of our paper.

Comment 1:

Improve the English language.

Answer 1:

We had a native check again.

Comment 2:

Change the number "2" in the title per "two".

Answer 2:

There is not an applicable point in the title.

Comment 3:

Improve the quality of your abstract.

Answer 3:

We had checked the abstract again.

Comment 4:

Provide the type of the study and the power of the sample size in the Materials and Methods section.

Answer 4:

This study was a cross-sectional study. In addition, we considered a variety of factors, such as workforce, when determining the study regions and number of participants. However, a priori sample size calculation was not performed for this study.

We added the information in the “Materials and Methods” section and “Discussion” section as the limitation.

Comment 5:

Figure 1 - Include the reasons for patients' loss.

Answer 5:

We included the reason for participant’s loss.

“No longer wanted to participate”.

Comment 6:

Did you consider the Skewness and kurtosis test for normality p>0.005? Is that correct?

Answer 6:

We are sorry. “p>0.005” was a mistake of “p>0.05”.

Comment 7:

I could not understand "Table 1" because it is not a table.

Answer 7:

We are sorry. “table” was a mistake of “figure”.

Comment 8:

The authors should explore the data obtained and write that in the results section. This section should be informative with a straightforward interpretation for the readers.

Answer 8:

We agreed with the reviewer’s comment. We modified the “Results” section.

Comment 9:

Please, do not write only "p." Change it for "p-value."

Answer 9:

According the reviewer’s suggestion, we changed to “p value”.

Comment 10:

Fix the Table 4.

Answer 10:

a longitudinal study of people as they developed periodontal disease showed that the

microbial dysbiosis occurred after disease developed.

7

 Animal studies showed that microbial

dysbiosis was reversed simply by resolution or control of the inflammation, consistent with the

model that inflammation was the driver of biofilm dysbiosis, and supporting the emerging data

suggesting that inflammation is the underlying cause of the disease.

8,9

a longitudinal study of people as they developed periodontal disease showed that the

microbial dysbiosis occurred after disease developed.

7

 Animal studies showed that microbial

dysbiosis was reversed simply by resolution or control of the inflammation, consistent with the

model that inflammation was the driver of biofilm dysbiosis, and supporting the emerging data

suggesting that inflammation is the underlying cause of the disease.

8,9

We modyfied the Table 4.

Comment 11:

Include the limitations of the present study in your discussion.

Answer 11:

We have added the limitation of the study in the “Discussion” section, as followes.

“The present study did have some limitations. First, We were not able to carry out detailed analysis such as improvement in physical fitness after the intervention due to lack of enough sample, and there was a lack of information about the subjects’ medical histories and diseases related to aging, according to the evidence based dentistry. Second, we considered a variety of factors, such as workforce, when determining the study regions and number of participants. However, a priori sample size calculation was not performed for this study. Third, we were not able to carry it out to measure of swallowing and mastication functions because a specialized measure such as endoscope or x-ray was necessary. Finally, we did not evaluate the participants’ occlusal status or prosthetic condition. A more detailed analysis is needed to confirm the relationship between oral function and physical fitness.”

Reviewer 2 Report

The authors aimed to evaluate the effects of an oral function training program on the improvement in physical performance induced by physical function training in elderly individuals.

The study covers some issues that have been overlooked in other similar topics. The structure of the manuscript appears adequate and well divided in the sub-paragraphs. Moreover, the study is easy to follow, but some issues should be improved. The manuscript needs moderate grammar correction. Please also check typos thorough the text.

Discussion section: The authors stated: "For example, we were not able to carry out detailed analysis such as improvement in physical fitness after the intervention due to lack of enough sample, and there was a lack of information about the subjects’ medical histories".

The statement could be improved with the following phrase and related references:

"For example, we were not able to carry out detailed analysis such as improvement in physical fitness after the intervention due to lack of enough sample, and there was a lack of information about the subjects’ medical histories and diseases related to aging, according to the evidence based dentistry." (please see DOI 10.7150/ijms.4.174   ;    DOI 10.1016/S0098-2997(96)00013-1).

Conclusions need to be increased with more related text and future insights on this matter.

Author Response

Our point-by-point reply to the reviewer 2’s comments

We are grateful to reviewer 2 for the critical comments and useful suggestions that have helped us to improve our paper considerably. As indicated in the followings, we have taken all these comments and suggestions into account in the revised version of our paper.

Comment 1:

The manuscript needs moderate grammar correction. Please also check typos thorough the text.

Answer 1:

The manuscript was checked by the native speakers.

Comment 2:

Discussion section: The authors stated: "For example, we were not able to carry out detailed analysis such as improvement in physical fitness after the intervention due to lack of enough sample, and there was a lack of information about the subjects’ medical histories".

The statement could be improved with the following phrase and related references:

"For example, we were not able to carry out detailed analysis such as improvement in physical fitness after the intervention due to lack of enough sample, and there was a lack of information about the subjects’ medical histories and diseases related to aging, according to the evidence based dentistry." (please see DOI 10.7150/ijms.4.174   ;    DOI 10.1016/S0098-2997(96)00013-1).

Answer 2:

According to the reviewer’s suggestion, we changed the sentences as follows.

“We were not able to carry out detailed analysis such as improvement in physical fitness after the intervention due to lack of enough sample, and there was a lack of information about the subjects’ medical histories and diseases related to aging, according to the evidence based dentistry.”

Comment 3:

Conclusions need to be increased with more related text and future insights on this matter.

Answer 3:

According to the reviewer’s suggestion we changed the conclusion.

Reviewer 3 Report

Dear authors, I have read your article with interest and some changes are needed. However, the increase in average age and quality of life certainly correlates also with the improvement in physical and oral health. therefore I congratulate the authors for the interesting topic, full of food for thought.

  • Line 4, the head number must be set before the comma. Please correct that.

  • Line 13, is not necessary all the informations of corresponding author. You should write correspondence: email,phone number and eventually fax. Please correct that

  • The abstract is well written but it should be maximum 200 words.

  • The introduction is too short and should be augmented with some references to show sufficient background of the topic. Important factors in assessing the activities of daily living are also muscle mass and muscle strength, which can strongly influence the quality of life of people. To enhance the significance of the article, it may be helpful to view this meta-analysis:  Wang DXM, Yao J, Zirek Y, Reijnierse EM, Maier AB. Muscle mass, strength, and physical performance predicting activities of daily living: a meta-analysis. J Cachexia Sarcopenia Muscle. 2020 Feb;11(1):3-25. doi: 10.1002/jcsm.12502. Epub 2019 Dec 1. PMID: 31788969; PMCID: PMC7015244.

  • Line 68, the authors affirms that “Few studies have evaluated the effects of combining physical function training with oral function training”. Where are the references? These should be placed in brakets.

  • What is the null hypothesis?

  • In recent years, good nutrition is considered the basis for achieving a good quality of life. It would be necessary to add some references of how a good diet, together with constant physical activity, can increase not only the quality of life but also the life expectancy of people. Please read and cite this article: Xu F, Cohen SA, Lofgren IE, Greene GW, Delmonico MJ, Greaney ML. Relationship between Diet Quality, Physical Activity and Health-Related Quality of Life in Older Adults: Findings from 2007-2014 National Health and Nutrition Examination Survey. J Nutr Health Aging. 2018;22(9):1072-1079. doi: 10.1007/s12603-018-1050-4. PMID: 30379305.

  • Lines 80-83, the authors affirm that the patients complete a basic health checklist provided by the Ministry of Health, Labour and Welfare. it is necessary to add the reference and it is strongly reccomended add an explanatory figure of the checklist.

  • Line 102, missing reference

  • Line 177, what type of digital counter? Please write the manufacturer on brakets as shown: ( Manufacturer, city, state)

  • it is not clear how the functionality of the chewing muscles was assessed

  • Line 255, the authors affirm that they do not take in consideration swallowing and mastication functions. what is the reason for this decision? Please explain that

  • Tables and images must be set following the indications of the Journal. Please correct them

  • At the end of the article the paragraph about the authors' contributions is missing. Please add it

Author Response

Our point-by-point reply to the reviewer 3’s comments

We are grateful to reviewer 3 for the critical comments and useful suggestions that have helped us to improve our paper considerably. As indicated in the followings, we have taken all these comments and suggestions into account in the revised version of our paper.

Comment 1:

Line 4, the head number must be set before the comma. Please correct that.

Answer 1:

We changed the location of the head number.

Comment 2:

Line 13, is not necessary all the informations of corresponding author. You should write correspondence: email,phone number and eventually fax. Please correct that

Answer 2:

We corrected the information of the corresponding author.

Comment 3:

The abstract is well written but it should be maximum 200 words.

Answer 3:

We corrected the abstract in the less than 200 characters.

Comment 4:

The introduction is too short and should be augmented with some references to show sufficient background of the topic. Important factors in assessing the activities of daily living are also muscle mass and muscle strength, which can strongly influence the quality of life of people. To enhance the significance of the article, it may be helpful to view this meta-analysis:  Wang DXM, Yao J, Zirek Y, Reijnierse EM, Maier AB. Muscle mass, strength, and physical performance predicting activities of daily living: a meta-analysis. J Cachexia Sarcopenia Muscle. 2020 Feb;11(1):3-25. doi: 10.1002/jcsm.12502. Epub 2019 Dec 1. PMID: 31788969; PMCID: PMC7015244.

Answer 4:

 According to the reviewer’s suggestion, we corrected the introduction based on the article.

Comment 5:

Line 68, the authors affirms that “Few studies have evaluated the effects of combining physical function training with oral function training”. Where are the references? These should be placed in brakets.

Answer 5:

There is not the investigation as far as we know it. We changed the sentence.

Comment 6:

What is the null hypothesis?

Answer 6:

It is hypothesized that the combination of oral function training to general physical training might give bigger improvement of not only oral function but also the physical performance.

Comment 7:

In recent years, good nutrition is considered the basis for achieving a good quality of life. It would be necessary to add some references of how a good diet, together with constant physical activity, can increase not only the quality of life but also the life expectancy of people. Please read and cite this article: Xu F, Cohen SA, Lofgren IE, Greene GW, Delmonico MJ, Greaney ML. Relationship between Diet Quality, Physical Activity and Health-Related Quality of Life in Older Adults: Findings from 2007-2014 National Health and Nutrition Examination Survey. J Nutr Health Aging. 2018;22(9):1072-1079. doi: 10.1007/s12603-018-1050-4. PMID: 30379305.

Answer 7:

According to the reviewer’s suggestion, we added the information based on the references.

Comment 8:

Lines 80-83, the authors affirm that the patients complete a basic health checklist provided by the Ministry of Health, Labour and Welfare. it is necessary to add the reference and it is strongly reccomended add an explanatory figure of the checklist.

Answer 8:

According to the reviewer’s suggestion, we added the explanatory figure of the checklist (The Ministry of Health, Labour and Welfare of Japan Home Page. Available online: https://www.mhlw.go.jp/topics/2009/05/dl/tp0501-1c_0001.pdf (Japanese)).

Comment 9:

Line 102, missing reference

Answer 9:

We added the reference.

Comment 10:

Line 177, what type of digital counter? Please write the manufacturer on brakets as shown: ( Manufacturer, city, state)

Answer 10:

a longitudinal study of people as they developed periodontal disease showed that the

microbial dysbiosis occurred after disease developed.

7

 Animal studies showed that microbial

dysbiosis was reversed simply by resolution or control of the inflammation, consistent with the

model that inflammation was the driver of biofilm dysbiosis, and supporting the emerging data

suggesting that inflammation is the underlying cause of the disease.

8,9

a longitudinal study of people as they developed periodontal disease showed that the

microbial dysbiosis occurred after disease developed.

7

 Animal studies showed that microbial

dysbiosis was reversed simply by resolution or control of the inflammation, consistent with the

model that inflammation was the driver of biofilm dysbiosis, and supporting the emerging data

suggesting that inflammation is the underlying cause of the disease.

8,9

We adde the manufacturer.

“Takei Scientific Instruments Co., Ltd. Niigata, Japan”

Comment 11:

It is not clear how the functionality of the chewing muscles was assessed.

Answer 11:

We have added the information about the assessment as follows.

“Two trained dentists evaluated OD and the number of remaining teeth in both the oral intervention and control groups at baseline and the 3-month follow-up examination. OD is a measure of orofacial motor skills. For the assessment of OD, the participants were asked to articulate the /pa/ syllable repeatedly as quickly as possible for 5 seconds, and the number of articulations was counted. The same procedure was repeated for the syllables /ta/ and /ka/, and the number of articulations was counted using a digital counter (Takei Scientific Instruments Co., Ltd. Niigata, Japan). OD scores were calculated separately for the /pa/, /ta/, and /ka/ syllables and recorded as the mean number of articulations per second. The ability to repeat the syllable /pa/ is a measure of lip function. On the other hand, the ability to repeat the syllables /ta/ and /ka/ is indicative of tongue root motor skills.”

Comment 12:

Line 255, the authors affirm that they do not take in consideration swallowing and mastication functions. what is the reason for this decision? Please explain that.

Answer 12:

We were not able to carry it out to measure of swallowing and mastication functions because a specialized measure such as a endoscope or x-ray was necessary.

We addeed the information as the limitation in the “Discussion” sectiuon.

Comment 13:

Tables and images must be set following the indications of the Journal. Please correct them.

Answer 13:

We correct the Tables and figures.

Comment 14:

At the end of the article the paragraph about the authors' contributions is missing. Please add it.

Answer 14:

We added the authors' contributions.

Author Contributions: M. S.: Idea of the study, Analysis and interpretation, Drafting of the article, Critical revision of the article for important intellectual content, A.Y.: Idea of the study, Analysis and interpretation, Drafting of the article, Critical revision of the article for important intellectual content, A. O.: Collection and assembly of data.

Round 2

Reviewer 1 Report

I’m satisfied with the modifications in the manuscript.